# Impact of Antithrombotic Regimen and Platelet Inhibition Extent on Leaflet Thrombosis Detected by Cardiac MDCT after Transcatheter Aortic Valve Replacement

**DOI:** 10.3390/jcm8040506

**Published:** 2019-04-12

**Authors:** Charline Jimenez, Mickaël Ohana, Benjamin Marchandot, Marion Kibler, Adrien Carmona, Marilou Peillex, Joe Heger, Antonin Trimaille, Kensuke Matsushita, Antje Reydel, Sébastien Hess, Laurence Jesel, Patrick Ohlmann, Olivier Morel

**Affiliations:** 1Université de Strasbourg, Pôle d’Activité Médico-Chirurgicale Cardio-Vasculaire, Nouvel Hôpital Civil, Centre Hospitalier Universitaire, 67000 Strasbourg, France; charline.jimenez@chru-strasbourg.fr (C.J.); benjamin.marchandot@chru-strasbourg.fr (B.M.); marion.kibler@chru-strasbourg.fr (M.K.); adrien.carmona@chru-strasbourg.fr (A.C.); marilou.peillex@chru-strasbourg.fr (M.P.); joe.heger@chru-strasbourg.fr (J.H.); antonin.trimaille@chru-strasbourg.fr (A.T.); matsuken_22@yahoo.co.jp (K.M.); anne-claire.reydel@chru-strasbourg.fr (A.R.); sebastien.hess@chru-strasbourg.fr (S.H.); Laurence.JESEL-MOREL@chru-strasbourg.fr (L.J.); Patrick.Ohlmann@chru-strasbourg.fr (P.O.); 2Université de Strasbourg, Département de Radiologie, Nouvel Hôpital Civil, Centre Hospitalier Universitaire, 67000 Strasbourg, France; mickael.ohana@chru-strasbourg.fr; 3UMR INSERM 1260 Regenerative Nanomedicine, Université de Strasbourg, 67000 Strasbourg, France

**Keywords:** TAVR, subclinical leaflet thrombosis, heart valve thrombosis, multidetector computed tomography, anticoagulation therapy, platelet, stroke, aortic valve stenosis, valve dysfunction

## Abstract

The impact of antithrombotic regimen and platelet inhibition extent on subclinical leaflet thrombosis (SLT) detected by cardiac multidetector computed tomography (MDCT) after transcatheter aortic valve replacement (TAVR) is not well established. Hypoattenuation affecting motion (HAM) has been proposed as a surrogate marker of SLT, and is characterized by hypoattenuated leaflet thickening (HALT) and concomitant reduction in leaflet motion (RELM). We sought to investigate (i) the prevalence of HAM and HALT after TAVR detected by MDCT, (ii) the predictors of SLT, (iii) the impact of oral anticoagulant (OAC) and platelet inhibition extent assessed by platelet reactivity index vasodilator stimulated phosphoprotein (PRI-VASP) and closure time adenosine diphosphate (CT-ADP) on SLT. Of 187 consecutive patients who underwent TAVR from 1 August 2017 to 31 March 2018, 90 of them had cardiac CT at relevant follow-up. Clinical, biological, echocardiographic, procedural characteristics and treatments were collected before, at discharge, and 1 year after TAVR. P2Y_12_ platelet inhibition extent and primary haemostasis disorders were investigated using platelet PRI-VASP and CT-ADP point-of-care assays. Eighty-five post-TAVR CTs out of 90 were ranked for clarity and assessed with sufficient diagnostic quality. HAM was evidenced in 13 patients (15.3%) and HALT in 30 patients (35%). Procedural characteristics, including aortic valve calcium score, annulus size, or procedural heparin regimens, were equivalent between groups. Likewise, no impact of P2Y_12_ inhibition (PRI-VASP) nor primary haemostasis disorders (CT-ADP) on SLT could be evidenced. No impact of SLT on valve deterioration evaluated by transthoracic echocardiography (TTE) and clinical events could be established at 12 months follow-up. By multivariate analysis, lack of oral anticoagulant therapy at discharge (HR 12.130 CI 95% (1.394–150.582); *p* = 0.028) and higher haemoglobin levels were evidenced as the sole independent predictors of SLT. In four patients with HAM, MDCT follow-up was obtained after initiation of OAC therapy and showed a complete regression of HAM. SLT was evidenced in a sizeable proportion of patients treated by TAVR and was mainly determined by the lack of oral anticoagulant therapy. Conversely, no impact of platelet inhibition extent on SLT could be evidenced.

## 1. Introduction

Transcatheter aortic valve replacement (TAVR) has become the standard of care in patients with severe symptomatic aortic valve stenosis and who are at intermediate or high surgical risk [1,2]. Despite recent advances, a wide range of ischemic and bleeding complications might mitigate the beneficial effect of TAVR procedures [3]. Although early stent valve thrombosis after TAVR remains a rare complication, recent reports have emphasized that subclinical leaflet thrombosis (SLT) is detected in a sizeable proportion of patients (7–15%) with normal echocardiographic parameters. Early strokes (<48 h) are mainly linked to periprocedural debris embolization and are unlikely to be targeted by antithrombotic regimen. The one-year stroke rate and thromboembolic complications incidence after TAVR vary from 2 to 3% [4]. Multidetector computed tomography (MDCT) has emerged as the gold standard in the TAVR prosthesis sizing assessment and more recently contrast-enhanced MDCT has emerged to detect post-TAVR asymptomatic leaflet thrombosis [5]. Hypoattenuation affecting motion (HAM) corresponds to SLT and is characterized by hypoattenuation leaflet thickening (HALT) and concomitant reduction in leaflet motion (RELM) evidenced by MDCT or by the elevation of the mean aortic transvalvular gradient ≥20 mmHg or increase more than 10 mmHg compared with baseline [6]. Up to now, the clinical repercussion of these imaging findings remains unclear. Recently, special attention was given to their eventual role in the increase of thromboembolic events but also on bioprosthetic valve dysfunction and durability. Given the frequency of incidental SLT, the adequacy of current antithrombotic/antiplatelet strategies has been questioned and various ongoing trials (ATLANTIS, POPular-TAVI, ENVISAGE-TAVI, AUREA, and AVATAR trials) will undoubtedly refine optimal strategies. Tailoring the antithrombotic therapy after TAVR is particularly challenging because the high-risk profile of this commonly elderly population leads to a significant overlap in the risk of both ischemic and bleeding events. To avoid ischemic complications, full-dose anticoagulation (usually intravenous heparin) is administered during the TAVR procedure and dual antiplatelet therapy (DAPT) by aspirin plus clopidogrel has been the recommended antithrombotic treatment, following the procedure [1,2] based on the empiric extrapolation of the percutaneous coronary intervention (PCI) experience. Current guidelines recommend the use of DAPT with aspirin and clopidogrel after TAVR to obviate the metallic stent mediated risk of thrombosis/embolization followed by long-term single antiplatelet therapy with aspirin alone. Oral anticoagulant’s (OAC) use has being restricted to patients with another indication of OAC—mainly atrial fibrillation. In the last two decades, several studies have emphasized that high platelet reactivity under P2Y_12_-inhibitors has been associated with increased risk of ischemic events including stent thrombosis and also cerebrovascular events [7]. By contrast, patients with extremely low platelet reactivity were demonstrated to be at increased risk of bleeding without any further benefit in stent thrombosis [8].

In the present study, we sought to investigate (i) the prevalence of HAM and HALT after TAVR detected by MDCT, (ii) predictors of SLT, and (iii) the impact of OAC and platelet inhibition extent assessed by platelet reactivity index vasodilator stimulated phosphoprotein (PRI-VASP) and closure time adenosine diphosphate (CT-ADP) on SLT.

## 2. Materials and Methods

### 2.1. Study Design and Patients

From 1 August 2017 to 31 March 2018, 187 consecutive patients underwent TAVR at our institution (Nouvel Hôpital Civil, Université de Strasbourg, France). The indication for TAVR and vascular access sites were assessed by the local heart team. The protocol was approved by the local ethics committee. All participants gave their informed written consent before the procedure and agreed to the anonymous processing of their data (France 2 Registry). Patients received aspirin (75 mg to 160 mg) and clopidogrel (300 mg for loading dose and 75 mg/day for maintenance dose) before TAVR, with ongoing DAPT after the procedure for 3 months. The loading of clopidogrel was not performed when the patient was under chronic clopidogrel therapy. In patients treated with OAC, clopidogrel was not administered, and OAC plus aspirin were continued for 3 months. Routinely, OAC therapy was discontinued 5 days prior to the procedure. CT imaging was planned at discharge. Of these patients, 90 had high-resolution cardiac CT performed at various times after TAVR. In the presence of HAM, OAC therapy was initiated and a new contrast MDCT was performed in the follow up to assess consistent regression of HAM images. Ninety-seven patients did not perform post-TAVR contrast MDCT due to death, severe impaired renal function (glomerular filtration rate ≤30 mL/min/1.73 m^2^) or loss of follow-up (Figure 1).

### 2.2. Study Devices and Procedures

For TAVR, the balloon-expandable Edwards SAPIEN XT or S3 prosthesis (Edwards Lifesciences, Irvine, CA, USA) and the self-expandable CoreValve or Evolut-R (Medtronic, Irvine, CA, USA) were used. During the procedure, 100 IU/kg of unfractionated heparin was administered to achieve an activated clotting time of 250 to 350 s. At the end of the procedure, heparin was antagonized with 100 IU/kg of protamine.

### 2.3. Blood Samples

Whole blood samples were collected the day before and 24 h after TAVR. Blood samples were immediately collected into a sodium citrate tube (VD Vacutainer® sodium citrate tubes) and sent to the haemostasis laboratory (EFS Alsace, France) for a platelet VASP phosphorylation analysis within 48 h. A standardized flow cytometric assay (Platelet VASP, Biocytex, Diagnostica Stago, Asnières, France) was used to assess VASP phosphorylation in all patients. VASP phosphorylation levels reflect P2Y_12_ inhibition and are expressed as the platelet reactivity index (PRI), calculated from the median fluorescence intensity (MFI) of samples incubated with prostaglandin E1 (PGE1) or PGE1 and ADP according to the formula PRI = ((MFIPGE1 − MFIPGE1 + ADP)/MFIPGE1) × 100. Patients were considered to have a low platelet response to clopidogrel (low-responder) if their PRI was >50%, and a normal response to clopidogrel (responder) if their PRI was ≤50% as previously described [9]. Analysis of CT-ADP with the primary haemostasis point-of-care assay PFA-100 (Siemens Healthcare) was performed according to the manufacturer’s recommendations. In the setting of TAVR, this point-of-care assay is mainly used as a surrogate marker of high molecular weight von Willebrand defect.

### 2.4. CT Acquisition Protocol

Pre- and post-TAVI ECG-gated MDCT examinations were performed using a second- or third- generation 320-row CT scanner (Aquilion ONE Vision Edition, Aquilion One Genesis, Canon Medical Systems, Japan). The aortic root CT angiography was acquired in volume mode using a retrospective ECG-gated acquisition and the following CT parameters: 16 cm width, 100 kV for pre-TAVI and 135 kV for post-TAVI, gantry rotation time of 0.275 s, auto-mA maxed at 300, acquisition over 1 heartbeat. Acquisition was obtained after a bolus injection of 50 to 70 mL of iomeprol 400 mg/mL (Iomeron®, Bracco, Italy), using an automatic power injector at a rate of 3.5 mL/s, followed by 30 mL of saline chaser at a rate of 3 mL/s. The acquisition was triggered using a bolus-tracking technique with a Region of Interest (ROI) positioned in the descending thoracic aorta and a 180 Hounsfield Units (HU) threshold. Aortic valve calcium score and aortic valve annulus sizing were determined on the pre-TAVR CT by one radiologist using Vitrea software in version 6.6 (Vital Imaging, Minnetonka, MN, USA).

CT imaging definition of SLT was defined according to previous definitions [6]. We first began to evaluate leaflet thickening (i.e., HALT) in diastole in two reconstructed planes. In the event of HALT, maximal leaflet thickening and the number of leaflets implicated were recorded. RELM was determined using cine reconstructions. Post-TAVR CT analysis was done in a consensus reading session with 1 experienced cardiac radiologist (MO, with 12 years of experience in CT) and 1 cardiologist (CJ, with 1 year of experience in CT) using a dedicated workstation (Vitrea version 6.6).

### 2.5. Echocardiography Assessment

Transthoracic echocardiography (TTE) was performed at baseline, discharge, 1 and 12 months follow-up at the local echocardiography laboratory (Nouvel Hôpital Civil, Université de Strasbourg, France). Prosthetic assessment was done following current recommendations and guidelines of the American Society of Echocardiography [10]. Left ventricle ejection fraction (LVEF), left ventricle dimensions, left atrial volume, mean aortic transvalvular gradient, permeability index, cardiac output, acceleration aortic time, and effective surface area were recorded. 

### 2.6. Blood Collection and Laboratory Assays

Whole blood samples were collected by venipuncture the day before and 24 and 72 h after TAVR and at discharge.

### 2.7. Collection of Data

All data, at baseline and during follow-up, were collected retrospectively according to the Valve Academic Research Consortium-2 criteria [11]. The primary endpoint of the study was to determine the prevalence of SLT (HAM ± postprocedural elevation in transvalvular gradient) according to anticoagulant and antiplatelet therapy. The secondary endpoint was to determine predictive factors of HAM. In addition, we investigated the prevalence of HALT. In the event of HAM, antithrombotic treatment (VKA or NOACs) was initiated and a new CT scan was performed few months later to assess consistent regression of HAM images and restored leaflet motion. 

### 2.8. Statistical Analysis

Categorical variables are expressed as numbers (%), and continuous variables are expressed as mean ± SD or median and interquartile values. Differences between the groups, consisting of HAM+ vs. HAM− and HALT+ vs. HALT− were assessed with χ^2^ tests for categorical variables. Unpaired Student’s *t*-test was used to analyse continuous variables that showed normal distributions, and the Mann–Whitney test was used to analyse continuous variables with skewed distributions. Univariate and multivariate analysis was performed to identify independent predictors of HAM+. Variables with a univariate *p* < 0.05 were considered for subsequent multivariate models. A *p*-value < 0.05 was considered significant. Statistical analysis was performed using SPSS 17.0 for Windows (SPSS Inc., Chicago, IL, USA).

## 3. Results

### 3.1. Study Cohort

Out of 187 patients who underwent TAVR, post-TAVR MDCT was obtained in 90 patients after a median interval of 114 (65–205) days. MDCT met the quality criteria to identify HAM in 85 patients and in 86 patients for HALT. The flow chart of the study is given in Figure 1. CT reconstruction showing a normal appearance of the aortic valve; HALT without RELM and HAM images are illustrated in Figure 2.

### 3.2. Hypoattenuation Affecting Motion

HAM was evidenced in 13 patients (15.3%). Among them, only four patients (30.7%) had altered transvalvular gradients (mean gradient ≥20 mmHg and/or a change in gradient ≥10 mmHg compared to baseline values).

Baseline clinical, echocardiographic, MDCT, procedural, and biological characteristics according to HAM stratification (HAM+ and HAM−) are displayed in Table 1 and Table 2. Patients were mainly women (67.1%), 82.1 ± 6.04 years old with a mean STS score of 4.3 ± 3.4%. The mean LVEF was 56% and the mean aortic gradient 44.8 ± 12.1 mmHg. CT aortic annulus area was similar in the two groups (495 ± 104 vs. 514 ± 154 mm^2^, *p* = 0.635), and the mean aortic valve calcium score was 2292 ± 1373 HU. TAVR was performed through transfemoral access in 79 (92.9%) patients and through transcarotid approach in six (7.1%) patients. There was no difference in heparin doses and activated clotting time (ACT) values during the procedures. Sapiens valves (71.8%) were mainly used in this study with similar rates of implantation among groups. Postprocedural evaluations of platelet inhibition by PRI-VASP (specific to P2Y_12_ inhibition) and occlusion time by CT-ADP point-of-care assay (sensitive not only to P2Y_12_ inhibition but also to several confounding factors, such as von Willebrand factor, platelet count, and haematocrit) were similar between groups. The proportion of HAM was 15.9% in Low Responder patients to clopidogrel (PRI-VASP > 50%) and 13.6% in Responder patients (*p* = 0.530). At hospital discharge, haemoglobin (Hb) and creatine levels were higher in HAM+ patients. 

Antithrombotic therapies at hospital discharge are summarized in Table 3. The prevalence of HAM was lower among patients under OAC (only 1 patient was on VKA (7.7%), *p* = 0.012) compared to those with DAPT (12 (92.3%) of 13, *p* = 0.005). By multivariable analysis, lack of OAC and higher Hb levels were the sole independent predictors of HAM occurrence. By contrast, no impact of PRI-VASP nor CT-ADP on HAM occurrence could be demonstrated (Table 4). The distribution of occlusion time, measured by the CT-ADP, and platelet inhibition by PRI-VASP, according to HAM, are represented Figure 3 and Figure 4.

In case of SLT, DAPT was switched to conventional anticoagulant therapy (VKA or NOACs). Among the 13 patients with SLT, follow-up MDCT could only be obtained in four patients, three to nine months later. In all four cases, CT showed a complete regression of HAM after completing anticoagulation therapy. 

Thirty-day post-TAVR echocardiograms and one-year follow-up are represented in Table 5. At one-month follow-up, the two subsets of patients showed similar mean aortic transvalvular gradients (10.9 ± 5.6 vs. 12.6 ± 6.1 mmHg; *p* = 0.327), indexed aortic valve areas (1.06 ± 0.3 vs. 1.1 ± 0.3 cm^2^/m^2^; *p* = 0.654), and aortic acceleration times (79 ± 20 vs. 80 ± 21 m/s; *p* = 0.80). 

Clinical follow-up was obtained in 70 patients. One-year mortality, major adverse cardiac events (MACE), and bleeding events were comparable between the two groups. Likewise, no sur-risk of stroke could be evidenced in the HAM+ group. Altogether, in this cohort of limited size, nonsignificant impact of HAM on adverse outcome could be established (Table 5). 

### 3.3. Hypoattenuation Leaflet Thickening

MDCT quality enabled HALT assessment in 86 patients. HALT without RELM were evidenced in 30 patients (35%). The clinical, echocardiographic, MDCT, biological, and procedural characteristics for the two groups (HALT+ and HALT−) are shown in Table 6. There was no significant difference between the two groups, especially in terms of OAC and antiplatelet therapy at discharge. 

Platelet inhibition assessed by PRI-VASP and closure time by CT-ADP did not differ significantly between patients with and without HALT (Figure 5 and Figure 6).

The one-year follow-up stratified by HALT is summarized Table 7. No significant differences among patient’s subset could be evidenced at one-year follow-up.

## 4. Discussion

### 4.1. Main Results

The current report, drawn from a cohort of 90 patients, is among the first study to specifically evaluate the impact of platelet inhibition extent on the occurrence of subclinical leaflet thrombosis detected by MDCT. The salient results of the present study are (1) subclinical leaflet thrombosis is observed in a sizeable proportion of TAVR patients (15%) after a median follow-up of 114 (65–205) days; (2) lack of oral anticoagulant together with elevated Hb levels were evidenced as independent predictors of subclinical leaflet thrombosis; (3) occurrence of subclinical leaflet thrombosis was independent of the extent of platelet inhibition, as measured by the PRI-VASP or the CT-ADP assay; (4) no impact of subclinical leaflet thrombosis on clinical events could be established. Altogether, our findings suggest the main importance of OAC therapy in the prevention of subclinical leaflet thrombosis independently of platelet inhibition extent. The prevalence of HAM reported in the present study (15%), is consistent with recent data from SAVORY and RESOLVE large observational registries in which TAVR reduced leaflet motion could be evidenced in 11.9% of the cohort [12]. In TAVR, important variations in the occurrence of SLT have been recently underlined, depending, in part, on the criteria used and the quality of MDCT acquisition. In the report by Sondergaard [13], HALT was evidenced in 38.1% and complete HAM in 20.2% after 140 days. By contrast, other investigators have reported lower HALT rates ranging from 4% (Leetmaa T, Circ Cardiovasc Interv. 2015 [14]), 7% (Hansson NC, JACC 2016 [15]), to 10.3% (Pache G, Eur Heart J. 2016 [16]). Of particular interest, recent data have underlined that leaflet thrombosis occurred more frequently in transcatheter bioprosthetic valves than surgical ones [12]. 

The study of mechanisms involved in thrombus formation during TAVR is far beyond the scope of the present study. Among various hypotheses, it is likely that the thrombus formation is the result of a complex interplay involving calcifications, native anatomy, hemodynamic, flow stasis, haemostatic factors, procedural factors, and valve type. Interaction between the prosthetic valve consisting in a metallic stent frame and three biologics leaflets, into a native or bioprosthetic (valve in valve) aortic valve, would create a new anatomic geometry characterized by modifications of shear stress regimen and turbulence. It is likely that the formation of a neosinus (region between the native and transcatheter aortic valve leaflet) favouring blood stasis would provide an ideal reservoir for thrombus formation. Moreover, exposure of procoagulant factors by the native valve (including tissue factor or procoagulant microparticles) may contribute to trigger thrombotic process. As pointed at by Midha et al. [17] supra-annular position of the neosinus may reduce thrombotic potential owing to the reduction of blood stasis. Another important factor relies on the under expansion of the prosthesis as underlined in Sapiens 3 valves or in autopsy studies [18]. Moreover, the possible contribution of valve injury mediated by balloon valvuloplasty or inflammatory response in thrombosis onset remains to be determined [19]. Others authors have suggested that the structure composition of the valves may play a differential role in thrombus formation. Whilst CoreValve porcine pericardium valve contains a nickel and titanium alloy, the Edward Sapien bovine pericardium valve has a polyethylene terephthalate skirt and cobalt chromium stent that could trigger allergic reaction, IgE antibody formation, and the coagulation cascade [20]. Finally, the contribution of paravalvular leak as a main determinant of platelet activation has recently be underlined [21]. 

As extensively demonstrated in PCI, we sought to investigate whether impaired platelet inhibition evaluated either by PRI-VASP or by CT-ADP could contribute to subclinical leaflet thrombosis. In TAVR, we have previously demonstrated that prolonged CT-ADP (>180 s) is not only a marker or paravalvular leakage (through enhanced proteolysis of high molecular weight von Willebrand factors) but also constitutes an integrate marker of enhanced periprocedural and late bleeding risk [22,23]. In our hand no impact of impaired platelet inhibition of HAM and HALT phenomenon could be established. This result is consistent with the recent study from Nührenberg et al. [24], which did not find significant association between the occurrence of HALT and impaired platelet inhibition assessed by ADP test. In the present study, the mean CT-ADP at discharge was 149 s which is comparable with data from Nührenberg and coworkers, but also with those from patients with coronary artery disease [7]. Altogether, our finding suggests that the development of bioprosthetic valve thrombosis is mainly platelet independent, consistent with a primary role of contact phase activation and more likely to be targeted by OAC.

Among various factors possibly involved in the development of SLT, converging evidences have highlighted a key role of anticoagulant therapy in the prevention or treatment of silent valve thrombosis. In the RESOLVE and SAVORY registries, subclinical leaflet thrombosis was less frequent among patients treated by OAC (4%) than patients receiving DAPT (15%). In this report, SLT resolved in all patients receiving anticoagulant either by VKA or NOACS, whereas it persisted in 91% of patients under antiplatelet therapy alone. Similarly, in our experience, HAM could only be evidenced in 7.7% patients under OAC. Multivariable analysis confirmed the independent association between lack of OAC at hospital discharge and HAM phenomenon. Accordingly, in a larger registry by Hansson [15] and coworkers, lack of warfarin treatment was also pointed out as an independent predictor of valve thrombosis. To date, important controversies remain on the impact of valve thrombosis on aortic valve pressure gradients, valve deterioration and adverse clinical outcomes. In the present report, mean aortic gradient at 30-day and one-year follow-up were not significantly different among groups. However, significant elevation of the mean aortic transvalvular gradient could be evidenced in 4 HAM patients (30.7%). Accordingly, in the report by Chakravarty [12], an increase in aortic valve gradient could be evidenced in 14% of patients with subclinical leaflet thrombosis. Other authors have emphasized the view that the treatment with OAC leads not only to regression of valve thickening but also on changes in transvalvular pressure gradients [25]. In the setting of TAVR, the question of valve durability remains of paramount importance. From a pathophysiological point of view, it is likely that endothelial damage and subsequent thrombosis may contribute to the infiltration of inflammatory cells within the valvular tissue leading to adverse remodeling and deterioration. Consistent with this paradigm, recent pathological analysis of 23 explanted transcatheter heart valves has underlined the existence of a time-dependent degeneration of heart valve consisting of thrombus formation, endothelial hyperplasia, fibrosis, tissue remodeling, proteinase expression, and calcification [26]. In line with this view, registry data from a cohort of 1521 TAVI patients has elegantly depicted that the rate of valve deterioration, as assessed by elevation in mean transvalvular gradient was 2.8% at one year following TAVR. Of paramount importance, the absence of anticoagulation therapy at discharge was evidenced as a key factor of valve deterioration, together with valve in valve procedure and the use of small 23-mm valve [27]. Although the contribution of confounding factors is difficult to delineate precisely in a prospective registry (OAC is mainly given in AF patients, AF could lead to gradient underestimation, and subsequently could underestimate valve deterioration in OAC receiving patients), those findings suggest that OAC are of paramount importance in the development of valve deterioration overt time. 

Another crucial issue relies on the clinical impact of valve thrombosis on thromboembolic events. The pioneering work by Chakravarty [12] has suggested that SLT was associated with increased rates of transient ischemic attack (TIA). Conversely, in a large registry comprising 754 patients (among them 120 patients with valve thrombosis), no differences in overall mortality and stroke/TIA rates could be demonstrated after a 406-day follow-up. However, it should be emphasized that rates of stroke/TIA were surprisingly extremely low (<2%) for a TAVR population [28], which probably reflects the German policy characterized by a more liberal use of TAVR implantation that includes lower risk patients. By contrast, insights from the US FDA MAUDE database has highlighted that leaflet thrombosis is associated with adverse outcomes including stroke, cardiogenic shock and death [29]. 

The definition of optimal antithrombotic therapies following TAVR remains a matter of important controversies. Whereas the frequency of subclinical leaflet thrombosis and the possible link between SLT and valve deterioration advocate for a liberal use of OAC, safety concerns and the assessment of bleeding events remain key elements when prescribing OAC. Several groups including ours have recently underlined the paramount importance of bleeding events that significantly outweighed ischemic concerns [23,24,25,26,27,28,29,30]. Of major importance, late bleeding is considered as a main determinant of adverse outcome in the frail TAVR population. Ongoing trials (ATLANTIS trial NCT02943785, POPular-TAVI trial NCT02247128, ENVISAGE-TAVI trial NCT02943785, AUREA trial NCT01642134, and AVATAR trial NCT02735902) will provide important insights on the efficacy but also safety profile of OAC including NOACS. However, alerting signal was very recently provided by the GALILEO trial which was prematurely halted. In this study, RIVAROXABAN, a Xa inhibitor was associated with greater risk of all-cause mortality, thromboembolic events, and bleeding in TAVR patients.

### 4.2. Study Limitations

Several limitations should be taken into account in the interpretation of the data: (i) HAM and HALT are considered as surrogate markers of SLT and pathological evidences for SLT could not be established; (ii) antithrombotic therapy was not randomized; (iii) CKD patients with eGFR < 30 mL/min/1.73 m^2^ were excluded from the analysis; (iv) timing between the two MDCT was not standardized; (v) follow-up was only obtained in 70 (82% patients); and (vi) given the limited size of the cohort, analysis should be interpreted with caution and the findings viewed as exploratory and hypothesis generating.

## 5. Conclusions

SLT is evidenced in a sizeable proportion of patients treated by TAVR and is mainly determined the lack of anticoagulant therapy. Conversely, no impact of platelet inhibition extent on SLT could be evidenced. Ongoing randomized prospective studies will provide important insights on the optimal antithrombotic strategy after TAVR.

## Figures and Tables

**Figure 1 jcm-08-00506-f001:**
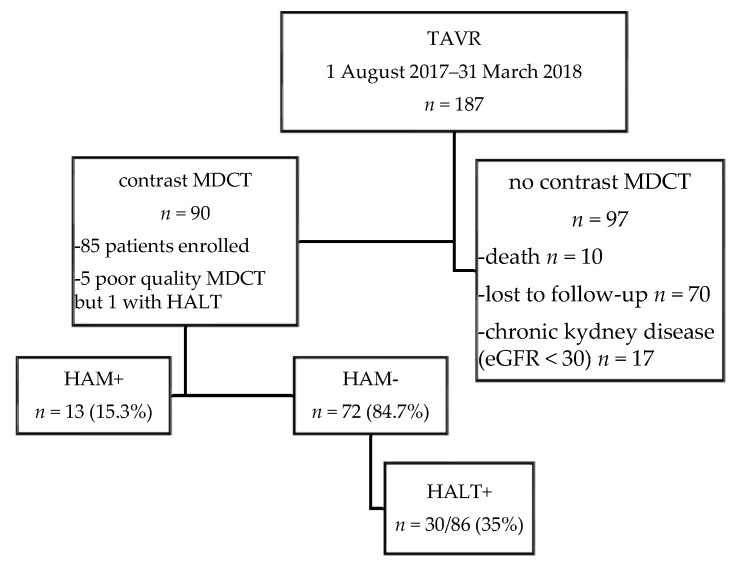
Flowchart of the study. Abbrevations: eGFR: estimated glomerular filtration rate; HAM: hypoattenution affecting motion; HALT: hypoattuated leaflet thickening; MDCT: multidetector computed tomography; TAVR: Transcatheter aortic valve replacement.

**Figure 2 jcm-08-00506-f002:**
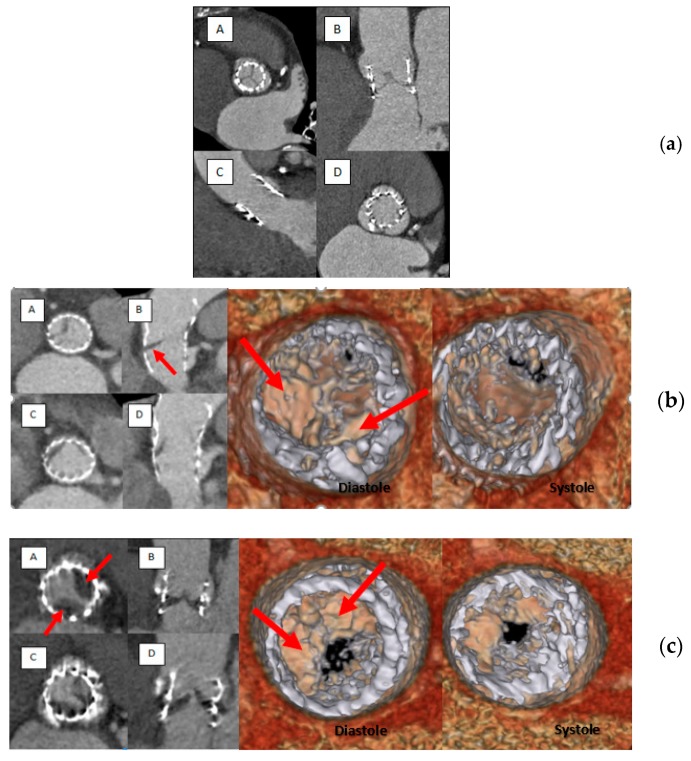
(**a**) MDCT images of normal (without HALT) Edward Sapien prosthesis in diastolic (short axis, A, and long axis, B) and in systolic (long axis, C, and short axis, D) phases. (**b**) MDCT images of HALT + (red arrow) RELM- in diastolic (short axis A, and long B axis) and in systolic (short axis, C, and long axis, D) phases. (**c**) MDCT images of HAM+ prosthesis: HALT+ (red arrow) and RELM + in diastolic (short axis, A, and long axis, B) and systolic (short axis, C, and long axis, D) phases.

**Figure 3 jcm-08-00506-f003:**
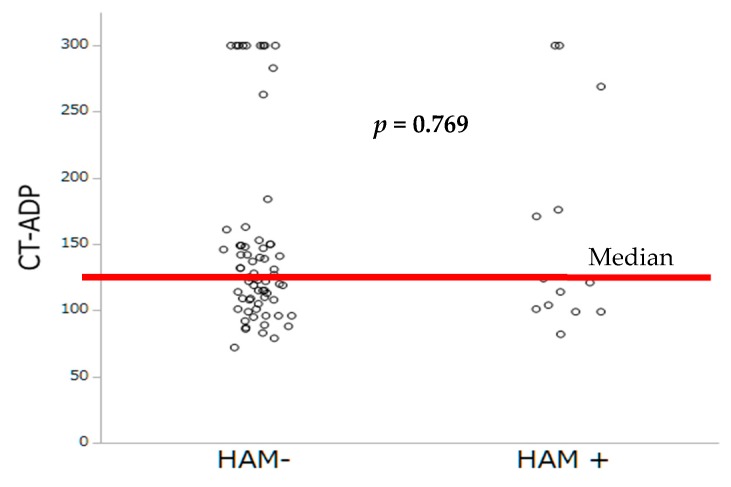
Occlusion time by CT-ADP according to the occurrence of HAM.

**Figure 4 jcm-08-00506-f004:**
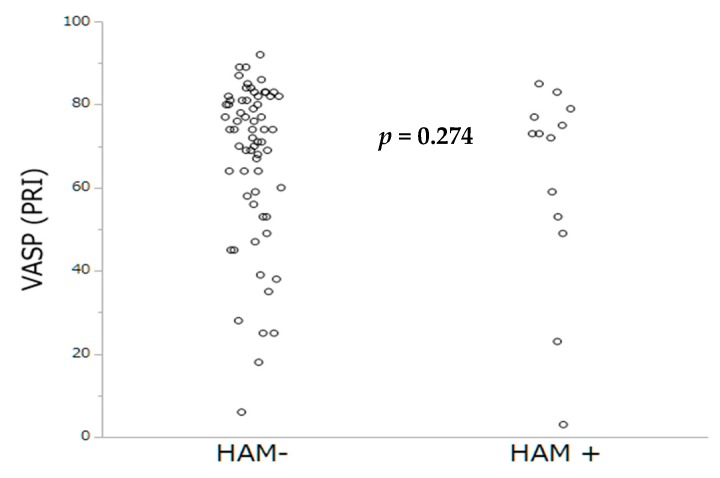
Platelet inhibition by PRI-VASP according to the occurrence of HAM.

**Figure 5 jcm-08-00506-f005:**
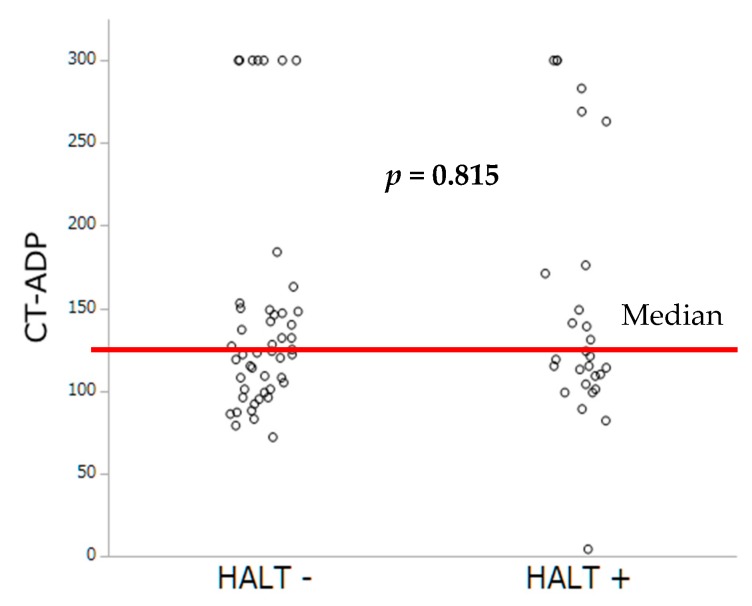
Occlusion time by CT-ADP according to the occurrence of HALT.

**Figure 6 jcm-08-00506-f006:**
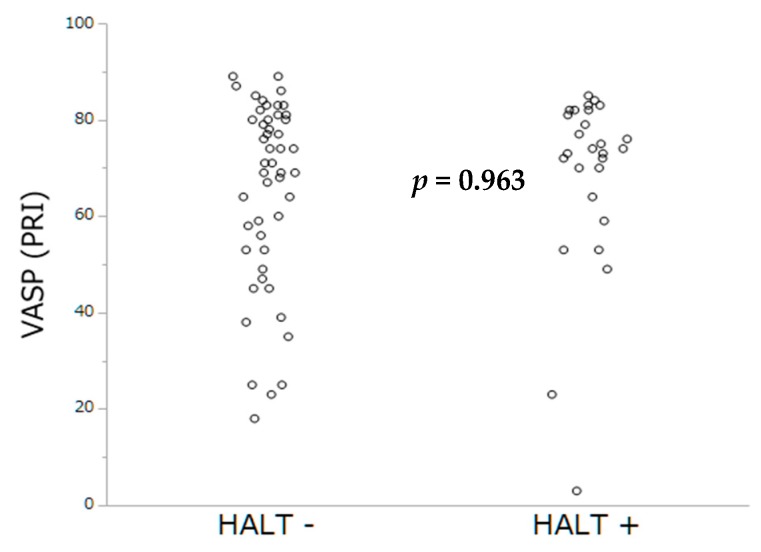
Occlusion time by VASP according to the occurrence of HALT.

**Table 1 jcm-08-00506-t001:** Baseline clinical, echocardiographic, and procedural characteristics according to HAM occurrence.

Characteristics	Total (*n* = 85)	HAM− (*n* = 72)	HAM+ (*n* = 13)	*p*-Value
Age-y	82.1 ± 6.04	82.1 ± 6.2	81.8 ± 5.6	0.919
Male sex-no (%)	28 (32.9)	23 (31.9)	5 (38.5)	0.645
BMI kg/m^2^	26.8 ± 6.1	27.2 ± 6.1	24.6 ± 5.8	0.15
EuroSCORE II, %	5.2 ± 4.3	5.5 ± 4.5	3.7 ± 2.5	0.168
STS score (%)	4.3 ± 3.4	4.3 ± 3.4	4.3 ± 3.4	0.829
AF-no (%)	19 (22.6)	18 (25)	1 (8.3)	0.201
CAD-no (%)	44 (53)	37 (52.9)	7 (53.8)	0.948
CKD-no (%)	45 (52)	39 (54.2)	6 (46.2)	0.594
Hypertension-no (%)	69 (81.2)	59 (81.9)	10 (76.9)	0.670
Current or former smoking history-no (%)	13 (15.3)	12 (16.7)	1 (7.7)	0.408
Dyslipidaemia-no (%)	53 (62.4)	45 (62.5)	8 (61.5)	0.947
Diabetes mellitus-no (%)	21 (24.7)	20 (27.8)	1 (7.7)	0.122
TTE preprocedural measurements				
LVEF, %	56.0 ± 13.2	55.9 ± 13.1	56.3 ± 14.7	0.934
LVEDD, mm	50.1 ± 8.3	50.1 ± 8.3	50.2 ± 8.9	0.965
AVAi, cm^2^/m^2^	0.44 ± 0.14	0.44 ± 0.1	0.44 ± 0.1	0.924
PI	0.2 ± 0.05	0.2 ± 0.1	0.2 ± 0.1	0.931
MA transvalvular gradient, mmHg	44.8 ± 12.1	44.8 ± 12.4	45 ± 10.2	0.698
LA volume, mL/m^2^	18.3 ± 17.3	48.4 ± 17.4	48 ± 17.6	0.956
SEVi, mL/m^2^	42.8 ± 13.3	42.6 ± 13.6	43.9 ± 12.8	0.736
CI mL/min/m^2^	3.2 ± 1.3	3.2 ± 1.4	2.9 ± 0.6	0.508
CT preprocedural measurements				
Aortic annulus area, mm^2^	499 ± 113	495 ± 104	514 ± 154	0.635
Aortic valve calcium score, HU	2292 ± 1373	3034 ± 1448	2797 ± 989	0.626
Procedural characteristics				
Approach				
Transfemoral-no (%)	79 (92.9)	67 (93.1)	12 (92.3)	0.923
Transcarotid-no (%)	6 (7.1)	5 (6.9)	1 (7.7)	0.923
Predilatation-no (%)	36 (42.4)	33 (45.8)	3 (23.1)	0.126
Valve in valve-no (%)	1 (1.2)	1 (1.4)	0 (0)	0.669
Valve				
Sapien-no (%)	61 (71.8)	50 (69.4)	11 (84.6)	0.263
CoreValve-no (%)	24 (28.2)	22 (30.6)	2 (15.4)	0.263
Size valve				
Sapien N 23 mm (%)	28 (32.9)	25 (34.7)	3 (23.1)	0.411
Sapien N 26 mm (%)	17 (20)	13 (18.1)	4 (30.8)	0.292
Sapien N 29 mm (%)	15 (17.6)	12 (16.7)	3 (23.1)	0.577
CoreValve N 23 mm (%)	2 (2.4)	2 (2.8)	0 (0)	0.543
CoreValve N 26 mm (%)	8 (9.4)	7 (9.7)	1 (7.7)	0.818
CoreValve N 29 mm (%)	10 (11.8)	9 (12.5)	1 (7.)	0.62
CoreValve N 31 mm (%)	1 (1.2)	1 (1.4)	0 (0)	0.669
CoreValve N 34 mm (%)	3 (3.5)	2 (2.8)	1 (7.7)	0.377
Total Procedure time, min	77.5 ± 26.2	78.2 ± 27.5	73.6 ± 17.1	0.593
Heparin, mg	232 ± 101	254 ± 110	111 ± 37	0.656
ACT, sec	277 ± 90	283 ± 94	240 ± 57	0.148

Abbreviations: AAT: aortic acceleration time; AF: atrial fibrillation; AVAi: indexed aortic valve area; BMI: body mass index: CAD: coronary artery disease; CKD: chronic kidney disease (creatinine > 150 µmol/L); CI: cardiac index; CT: computed tomography; HAM: hypoattenuation affecting motion; HU: Hounsfield units; LA: left atrial; LVEDD: left ventricle end diastole diameter; LVEF: left ventricle ejection function; MA: mean aortic; PI: permeability index; SEVi: stroke ejection volume index; TTE: transthoracic echocardiography.

**Table 2 jcm-08-00506-t002:** Biological characteristics.

Characteristics	Total (*n* = 85)	HAM− (*n* = 72)	HAM+ (*n* = 13)	*p*-Value
Hb, g/dL				
Baseline	12.4 ± 2.2	12.2 ± 2.0	13.0 ± 1.6	0.245
Post-TAVR	10.14 ± 1.6	9.9 ± 1.5	11.1 ± 1.3	0.015
Creatinine, µmol/L				
Baseline	113.9 ± 131.5	112.7 ± 133.9	121.2 ± 121.5	0.831
At discharge	104.7 ± 109.3	94.5 ± 74.3	160 ± 216.5	0.043
CT-ADP, s				
Baseline	221.2 ± 77.7	218.6 ± 76.9	235.8 ± 86.4	0.499
Post-TAVR	149.1 ± 73	147.2 ± 72.0	158.5 ± 79.8	0.769
VASP, %				
Post-TAVR	66.9 ± 18.4	68.0 ± 16.9	61.8 ± 24.6	0.274
PRI-VASP > 50%	63 (82.9)	53 (84.1)	10 (76.9)	0.530

Abbreviations: CT-ADP: closure time adenosine diphosphate; HAM: hypoattenuation affecting motion; Hb: Haemoglobin; PRI-VASP: platelet reactivity index vasodilator stimulated protein; TAVR: transcatheter valve replacement.

**Table 3 jcm-08-00506-t003:** HAM according to antithrombotic regimens.

Antithrombotic Regimens	Total (*n* = 85)	HAM− (*n* = 72)	HAM+ (*n* = 13)	*p*-Value
Antithrombotic treatment before TAVR				
Antiplatelet therapy-no (%)	42 (49.4)	33 (45.8)	9 (69.2)	0.12
OAC-no (%)	29 (34.1)	28 (38.9)	1 (7.7)	0.029
Antithrombotic treatment at discharge				
DAPT-no (%)	48 (56.5)	36 (50)	12 (92.3)	0.005
OAC-no (%)	33 (38.8)	32 (44.4)	1 (7.7)	0.012

Abbreviations: DAPT: dual antiplatelet therapy; HAM: hypoattenuation affecting motion; OAC: oral anticoagulant; TAVR: transcatheter aortic valve replacement.

**Table 4 jcm-08-00506-t004:** Predictors of HAM.

Characteristics	Univariate Analysis			Multivariate Analysis		
	HR	95% CI	*p*-Value	HR	95% CI	*p*-Value
Age	0.995	0.903–1.096	0.917			
Male sex	1.332	0.392–4.520	1.332			
EuroSCORE II	0.829	0.646–1.063	0.829			
Diabetes mellitus	0.217	0.026–1.777	0.154			
BMI	0.906	0.795–1.032	0.138			
Sapien valve	2.420	0.495–11.842	0.275			
Total procedure time	0.992	0.965–1.020	0.589			
UFH	0.993	0.973–1.014	0.526			
Predilatation	0.355	0.090–1.397	0.138			
Lack of OAC at discharge	10.154	1.253–82.528	0.030	12.130	1.394–150.582	0.028
Creatinine at discharge	1.004	0.999–1.008	0.092			
CT-ADP	1.002	0.994–1.010	0.611			
PRI-VASP > 50%	0.629	0.147–2.699	2.699			
Hb at discharge	1.730	1.101–2.719	0.017	1.887	1.148–3.104	0.012

Abbreviations: BMI: body mass index; CI: confidence interval; CT-APD: closure time adenosine diphosphate; HAM: hypoattenuation affecting motion; Hb: haemoglobin; HR: hazard ratio; OAC: oral anticoagulant; PRI-VASP: platelet reactivity index-vasodilator stimulated; UFH: unfractionated heparin.

**Table 5 jcm-08-00506-t005:** Thirty-day TTE evaluation and one-year follow-up endpoints according to HAM occurrence.

Follow-Up	Total (*n* = 70)	HAM− (*n* = 58)	HAM+ (*n* = 12)	*p*-Value
One-year follow-up endpoints				
Mortality-no (%)	1 (1.4)	1 (1.7)	0 (0)	0.647
MACE+ Bleeding complications-no (%)	6 (8.7)	4 (7.0)	2 (16.7)	0.281
Stroke and TIA-no (%)	4 (5.8)	3 (5.3)	1 (8.3)	0.679
Bleeding-no (%)	3 (4.4)	2 (3.5)	1 (9.1)	0.409
Rehospitalization-no (%)	32 (45.7)	25 (43.1)	7 (58.3)	0.335
Heart failure-no (%)	9 (12.9)	8 (13.8)	1 (8.3)	0.607
AF-no (%)	4 (5.8)	3 (5.3)	1 (8.3)	0.679
30-Day post-TAVR TTE				
LVEF, %	58.1 ± 10.3	58.2 ± 10.6	57.7 ± 8.9	0.879
MA transvalvular gradient, mmHg	11.2 ± 5.7	10.9 ± 5.6	12.6 ± 6.1	0.327
PI	0.54 ± 0.2	0.54 ± 0.1	0.49 ± 0.2	0.262
AVAi, cm^2^/m^2^	1.06 ± 0.3	1.06 ± 0.3	1.1 ± 0.4	0.654
AAT, m/s	79 ± 20	79 ± 20	80 ± 21	0.800
SEVi, mL/m^2^	46.4 ± 13	45.6 ± 13	51.2 ± 10	0.145
CI mL/min/m^2^	3.6 ± 1.5	3.7 ± 1.6	3.4 ± 0.9	0.603
1-Yeat post-TAVR TTE				
LVEF, %	58.6 ± 8.6	58.1 ± 9.0	60.7 ± 6.6	0.432
MA transvalvular gradient, mmHg	10.9 ± 5.5	10.8 ± 5.7	11.5 ± 5.0	0.75
PI	0.53 ± 0.2	0.53 ± 0.1	0.55 ± 0.2	0.643
AVAi, cm^2^/m^2^	0.99 ± 0.3	0.98 ± 0.3	1.06 ± 0.5	0.79
AAT, m/s	77.9 ± 15.5	77.7 ± 15.7	79.6 ± 15.7	0.805
SEVi, mL/m^2^	43.9 ± 10.0	44.3 ± 10.0	42.5 ± 10.9	0.702
CI mL/min/m^2^	3.2 ± 1.1	2.9 ± 1.0	3.9 ± 1.1	0.045

Abbreviations: AAT: aortic acceleration time; AF: atrial fibrillation; AVAi: indexed aortic valve area; CI: cardiac index; HAM: hypoattenuation affecting motion; LVEF: left ventricle ejection function; MA: mean aortic; MACE: major adverse cardiac events; TAVR: transcatheter aortic valve replacement; TIA: transient ischemic attack; TTE: transthoracic echocardiography; PI: permeability index; SEVi: stroke ejection volume index.

**Table 6 jcm-08-00506-t006:** Clinical baseline, echocardiographic, biological, and procedural characteristics for HALT.

Characteristics	Total (*n* = 86)	HALT− (*n* = 56)	HALT+ (*n* = 30)	*p*-Value
Age- y	82.1 ± 6.02	81.9 ± 6.7	82.5 ± 4.5	0.651
Male sex-no (%)	28 (32.6)	16 (28.6)	12 (40)	0.281
BMI kg/m^2^	26.8 ± 6.1	27.6 ± 6.2	25.2 ± 5.6	0.083
EuroSCORE II, %	5.5 ± 4.7	5.7 ± 4.7	4.9 ± 4.8	0.47
AF-no (%)	19 (22.4)	13 (23.2)	6 (20.7)	0.791
CAD-no (%)	45 (53.6)	29 (52.7)	16 (55.2)	0.831
CKD-no (%)	46 (53.5)	31 (55.4)	15 (50)	0.635
Hypertension-no (%)	70 (81.4)	46 (82.1)	24 (80)	0.808
Current or former smoking history-no (%)	13 (15.1)	10 (17.9)	3 (10)	0.332
Dyslipidaemia-no (%)	54 (62.8)	35 (62.5)	19 (63.3)	0.939
Diabetes mellitus-no (%)	21 (24.4)	15 (26.8)	6 (20)	0.485
TTE preprocedural measurements				
LVEF, %	56.13	55.9 ± 12.7	56.5 ± 14.2	0.864
LVEDD, mm	50.1 ± 8.3	49.6 ± 8.7	51.1 ± 7.4	0.452
AVAi, cm^2^/m^2^	0.44 ± 0.1	0.44 ± 0.2	0.44 ± 0.1	0.882
PI	0.2 ± 0.05	0.2 ± 0.05	0.19 ± 0.04	0.434
MA transvalvular gradient, mmHg	44.8 ± 12.0	44.6 ± 13.0	45.1 ± 10.1	0.861
LA volume, mL/m^2^	48.6 ± 17.3	47.1 ± 17.9	51.3 ± 16.4	0.321
SEVi, mL/m^2^	42.9 ± 13.3	43.4 ± 13.7	42.1 ± 12.8	0.687
CI mL/min/m^2^	3.2 ± 1.2	3.2 ± 1.1	3.2 ± 1.5	0.793
CT preprocedural measurements				
Aortic annulus area, mm^2^	498.3 ± 112	494.6 ± 98	502.8 ± 123	0.790
Aortic valve calcium score, HU	2991 ± 1376	3126 ± 1608	2513 ± 981	0.404
Procedural characteristics				
Approach				
Transfemoral-no (%)	80 (93)	51 (91.1)	29 (96.7)	0.332
Transcarotid- no (%)	6 (7)	5 (8.9)	1 (3.3)	0.332
Valve in valve procedure-no (%)	2 (2.3)	1 (1.8)	1 (3.3)	0.65
Valve				
Sapien-no (%)	61 (70.9)	38 (67.9)	23 (76.7)	0.391
CoreValve-no (%)	25 (29.1)	18 (32.1)	7 (23.3)	0.391
Size valve				
Sapien N 23 mm (%)	28 (32.6)	21 (37.5)	7 (23.3)	0.181
Sapien N 26 mm (%)	17 (19.8)	9 (16.1)	8 (26.7)	0.24
Sapien N 29 mm (%)	15 (17.4)	8 (14.3)	7 (23.3)	0.292
CoreValve N 23 mm (%)	3 (3.5)	2 (3.6)	1 (3.3)	0.954
CoreValve N 26 mm (%)	8 (9.3)	5 (8.9)	3 (10)	0.87
CoreValve N 29 mm (%)	10 (11.6)	7 (12.5)	3 (10)	0.73
CoreValve N 31 mm (%)	1 (1.2)	1 (1.8)	0 (0)	0.462
CoreValve N 34 mm (%)	3 (3.5)	2 (3.6)	1 (3.3)	0.954

Abbreviations: AAT: aortic acceleration time; AF: atrial fibrillation; AVAi: indexed aortic valve area; BMI: body mass index; CAD: coronary artery disease; CKD: chronic kidney disease (creatinine >150 µmol/L); CI: cardiac index; CT: computed tomography; HALT: hypoattenuated leaflet thickening; HU: Hounsfield units; LA: left atrial; LVEDD: left ventricle end diastole diameter; LVEF: left ventricle ejection function; MA: mean aortic; PI: permeability index; SEVi: stroke ejection volume index; TTE: transthoracic echocardiography.

**Table 7 jcm-08-00506-t007:** HALT at 30-day and 1-year follow-up.

Follow-Up	Total (*n* = 70)	HALT− (*n* = 46)	HALT+ (*n* = 24)	*p*-Value
1-Year follow-up endpoints				
Mortality-no (%)	1 (1.4)	1 (2.1)	0 (0)	0.471
MACE + Bleeding complications-no (%)	7 (10)	4 (8.7)	3 (12.5)	0.568
Stroke and TIA-no (%)	4 (5.7)	3 (6.5)	1 (4.2)	0.697
Bleeding-no (%)	4 (5.8)	1 (2.2)	3 (13)	0.015
Rehospitalization-no (%)	33 (46.5)	18 (39.1)	15 (60)	0.121
Heart failure-no (%)	10 (14.1)	6 (13)	4 (16)	0.777
AF-no (%)	5 (7.1)	2 (4.4)	3 (12)	0.089
30-Day post-TAVR TTE				
LVEF, %	58.2 ± 10.3	58.6 ± 10.8	57.4 ± 9.5	0.598
MA transvalvular gradient, mmHg	11.1 ± 5.5	10.9 ± 5.4	11.6 ± 5.7	0.57
PI	0.54 ± 0.15	0.56 ± 0.15	0.50 ± 0.15	0.08
AVAi, cm^2^/m^2^	1.06 ± 0.3	1.06 ± 0.3	1.05 ± 0.4	0.878
AAT, m/s	79.1 ± 19.9	79.9 ± 20.7	77.4 ± 18.6	0.606
SEVi, mL/m^2^	46.2 ± 13.2	45.3 ± 13.2	48.01 ± 12.8	0.402
CI mL/min/m^2^	3.6 ± 1.5	3.7 ± 1.6	3.4 ± 1.3	0.553
1-Year post-TAVR TTE				
LVEF, %	58.6 ± 8.6	57.7 ± 9.7	60.2 ± 5.9	0.367
MA transvalvular gradient, mmHg	10.9 ± 5.5	11 ± 5.8	10.8 ± 5.1	0.911
PI	0.53 ± 0.14	0.52 ± 0.14	0.55 ± 0.17	0.653
AVAi, cm^2^/m^2^	0.99 ± 0.34	0.98 ± 0.34	1.01 ± 0.35	0.832
AAT, m/s	77.9 ± 15.5	76.7 ± 15.5	80.8 ± 15.6	0.46
SEVi, mL/m^2^	43.9 ± 10	49 ± 9.5	41 ± 10.5	0.224
CI mL/min/m^2^	3.2 ± 1.1	3.1 ±1	3.2 ± 1.1	0.801

Abbreviations: AAT: aortic acceleration time; AF: atrial fibrillation; AVAi: indexed aortic valve area; CI: cardiac index; HALT: hypoattenuated leaflet thickening; LVEF: left ventricle ejection function; MA: mean aortic; MACE: major adverse cardiac events; TIA: transient ischemic attack; TTE: transthoracic echocardiography; PI: permeability index; SEVi: stroke ejection volume index.

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
