# Peer review of "Impact of Antithrombotic Regimen and Platelet Inhibition Extent on Leaflet Thrombosis Detected by Cardiac MDCT after Transcatheter Aortic Valve Replacement"

_jcm, 2019, doi:10.3390/jcm8040506_

Reviewer 1 Report

In this manuscript, the authors investigated the impact of anti-thrombotic regimen and platelet inhibition extent on SLT. They found that SLT is present in quite large proportion of patient treated by TAVR, and it is mainly determined the lack of anticoagulant therapy. This study includes 187 patients who underwent TAVR. The analysis is complemented, and the conclusion can be supported by the results. I have no major concerns.

Author Response

We would like to thank the Reviewer for his/her positive comment. Spelling errors were corrected throughout the manuscript

Reviewer 2 Report

Detailed characteristics of the study group, well-chosen methodology to achieve the intended goal and a practical approach to the described medical problem, indicate the high quality of the reviewed manuscript. 

Author Response

We would like to thank the Reviewer for his/her positive comment

Reviewer 3 Report

This is a really novel study focused on predictors of subclinical leaflet thrombosis (SLT) after TAVI. The authors have identified the lack of OAC therapy at discharge, but not platelet reactivity, as a predictor of future SLT. This is a relevant contribution into the on-going discussion about optimal post-TAVI medical therapy, which is important for the clinicans.  

I have only a minor comment: the authors might indicate which covariates were entered into multivariate regression as potential confounders when comparing HAM-positive and HAM-negative patients. In particular, did they enter only the covariates for which there was a significant difference between the both subgroups (HAM(+) vs. HAM(-), such as creatinine at discharge, or perhaps they also adjusted for some other potential confounders. 

Author Response

We would like to thank the Reviewer for his/her positive comment. In this study, variables with a univariate p<0.05 were considered for subsequent multivariate models. Therefore, creatinine at discharge (HR 1.004 95% CI (0.999-1.008) p = 0.092) was not enter into the multivariable analysis.

Changes: To take into account the Reviewer’s concern, the statistical analysis paragraph was modified with the addition of the sentence: variables with a univariate p<0.05 were considered for subsequent multivariate models

Reviewer 4 Report

The paper “Impact of Anti-thrombotic Regimen and Platelet Inhibition Extent on Leaflet Thrombosis detected by Cardiac MDCT after Transcatheter Aortic Valve Replacement” is an interesting study that reports on new results regarding the prevalence of HAM and HALT after TAVR detected by MDCT, predictive factors of Subclinical Leaflet Thrombosis (SLT), impact of oral anticoagulant (OAC) and platelet inhibition. The authors revealed that “HAM was evidenced in 13 patients (15.3%) and HALT in 30 (35%)” which is reasonable close to 17 (20.2%) and 32 (38.1%) as reported previously by Sondergaard et al, 2017 in a different cohort. These results provide additional supportive data both in terms of the prevalence rate as well as in terms of the lack of anticoagulant therapy. The authors have also not shied off putting most of the study limitations. In my mind, this study although fail to open new horizons for understanding mechanisms, but it does add significant information to the clinical data related to TAVR which is important. In general, the paper is well written, clearly formulated and easy to read, the Figures illustrate the text enough and help to follow the results. It also cover recent relevant references.

It do need minor English/spelling corrections, for example: in ‘Introduction’ section:

(ATLANTIS, POPular-TAVI, ENVISAGE-TAVI, AUREA and AVATAR trials) are one?? the way to refine optimal strategies.

Author Response

(The authors gave the same response as above.)
